# Nurturing the Positive Mental Health of Autistic Children, Adolescents and Adults alongside That of Their Family Care-Givers: A Review of Reviews

**DOI:** 10.3390/brainsci13121645

**Published:** 2023-11-27

**Authors:** Roy McConkey

**Affiliations:** Institute of Nursing and Health Research, Ulster University, Belfast BT15 1AP, UK; r.mcconkey@ulster.ac.uk

**Keywords:** autism, autistic, mental health, emotional wellbeing, family care-givers, low- and middle-income countries, review, international

## Abstract

The rising prevalence of autism internationally has been accompanied by an increased appreciation of the poorer mental health experienced by people with this condition and also of their family care-givers. In particular, higher incidences of anxiety and depression are reported in high-income nations and these conditions are likely to be under-recognised and under-reported in lower-resourced regions or countries. Mainstream mental health services seem to be ill-equipped to respond adequately to the needs of autistic persons and their care-givers. This literature review of 29 recently published reviews covering nearly 1000 journal articles summarises the insights and strategies that have been shown to promote the mental health and emotional wellbeing of autistic persons. In particular, a focus on family-centred, community-based supports is recommended that aim to enhance social communication, extend social connections and promote an individual’s self-esteem, self-determination and social motivation. These low-cost interventions are especially pertinent in low-resourced settings, but they can be used internationally to prevent mental illness and assist in the treatment of anxiety and depression in autistic persons and their family carers. The priority is to focus on primary-care responses with cross-sectoral working rather than investing in high-cost psychiatric provision.

## 1. Introduction

This review aims to summarise the existing knowledge about the mental health and emotional wellbeing of people with autism world-wide and especially how mental ill-health could be prevented. Fortuitously, there is a growing research base on which the planning and delivery of support services can be based. Moreover, the research studies have often been collated by investigators who conducted systematic reviews and meta-analyses of published articles from across the globe—although predominantly from high-income countries—that bring together the findings from individual studies. In all, 29 systematic reviews were identified that addressed specific topics related to the theme of this paper.

A literature review of the reviews was then undertaken with the primary aim of identifying recommendations for practice based on the insights and evidence across the diverse topics affecting the mental health of persons with autism. As you will read, the three main themes related to the aim of this paper were first identified and these form the main sections of this paper. Within each section, review papers published mostly within the last five years in an international, peer-reviewed journal were identified mainly through Google Scholar and cross-checked with other search engines such as Web of Science using the terms (or variants) “autism”, “mental health” and “review”. In addition, the reviews were augmented with pertinent individual papers to elaborate the issues identified in the reviews. Table A1 in Appendix A gives details about the selected review papers, all of which are listed in the references with a doi number provided to facilitate access to each review paper. In total, nearly 1000 studies were included in the identified reviews involving at least 275,000 people with autism and family carers. In this paper, the main conclusions from the reviews are reported along with a synthesis of the implications for practice. In those review papers that assessed the quality of the research studies, it was generally rated as low. Therefore, the findings have to be treated with a degree of caution, as their findings may not be replicable. Moreover, this literature review could usefully guide future scoping or systematic reviews on the topic of mental health and autism, which would help to overcome any subjective biases of this author [1]. Two further considerations were uppermost in the selection of reviews. The present paper aims to make practitioners and autism self-advocates more knowledgeable about the mental health of people with autism and how they could best be helped, rather than identifying further research topics, although that would be advantageous. Secondly, low- and middle-income countries need low-cost interventions rather than ones requiring teams of highly trained clinicians. But, that said, service commissioners in more affluent countries may also want to explore similar options given the financial pressures on current mental health services.

## 2. What Do We Know about Autism?

Autism is a life-long condition that affects the citizens of every nation. Unlike other widely known disabilities—physical, sensorial and cognitive—it was only in 1980 that autism was clinically recognised as a distinct condition. Disputes continued around its identification, but a consensus has emerged as to its essential features: [2] “*Persistent deficits in social communication and social interaction across multiple contexts” and “restricted, repetitive patterns of behavior, interests, or activities*”. Three levels of severity are proposed: “*Level 3—requires very substantial support, Level 2—Requires substantial support, and Level 1—requires support*”. These levels reflect the variation in the extent to which individuals experience autistic symptoms. Three further conditions also need to be met: (1) “*Symptoms must be present in the early developmental period (although they be masked in the earlier years)*”; (2) “*Symptoms cause clinically significant impairment in social, occupational, or other important areas of current functioning”.* (3) “*These disturbances are not better explained by intellectual disability (intellectual developmental disorder) or global developmental delay*”.

A recent meta-analysis of prevalence rates of the condition reported in 77 studies undertaken in 34 countries since 2012 [3] identified a median prevalence rate of one in 100 persons, but this varied widely across studies from one in 1000 to 4.4 per 100 persons. Most studies were conducted in the Americas and Europe, which reported much higher rates (median 1.3 and 1 per 100, respectively) than studies in South-East Asia (median 0.23 per 100) or the Western Pacific (0.28 per 100), a four-fold difference. There has been much speculation around the reasons for this, including variations in public awareness, differences in assessment and diagnostic services, the widening of the criteria for autism and the tools used to screen and assess for autism.

Internationally, these rates also reflect higher numbers of children being identified in recent years compared to earlier studies, but also the number of adult persons with this condition. The latter increase results from children growing older, and also because some are now receiving a diagnosis in adulthood despite it being undiagnosed when they were children. Nevertheless, many more adult persons remain undiagnosed or mis-diagnosed. In the USA, an estimated 2.2% (95% simulation interval (SI) of between 1.95 and 2.45%) or around 5.5 million US adults aged 18 to 84 could be living with some form of autism [4].

Autism can also occur alongside other conditions. For example, around one-quarter of persons in population studies are estimated to also have an intellectual disability, which too can vary in its severity from mild to severe intellectual disability [5]. Likewise, they may have hearing and vision problems, but an issue of growing concern is the presence of mental health difficulties in children, youth and adults with autism.

### 2.1. Mental Health and Autism

An “umbrella” review of comorbid psychiatric disorders among people with autism based on 14 systematic reviews and 12 meta-analysis articles published mostly from 2015 onwards in peer-reviewed journals, found that the prevalence of at least one psychiatric disorder was 54.8% (95% CI: 46.6–62.7%) [6]. However, the prevalence rates varied widely across the various reviews and meta-analyses, likely due to the characteristics of the people studied and the mental health assessments used.

One systematic review and meta-analysis that was undertaken of 96 international studies examined the presence of eight categories of co-occurring mental health problems in various samples of persons identified as having autism (range 23 to 36,947 cases) [7]. Comparisons could also be made with the presence of these conditions in comparable general or “neurotypical” samples. The top five conditions identified in people with autism are shown in Table 1. As you will read, the prevalences were significantly higher than their occurrence in the ”neurotypical” population.

The other less-frequently occurring conditions were bi-polar and related disorders, schizophrenia spectrum and psychotic disorders and sleep–wake disorders.

In sum, children and adults with autism are at a greater risk of experiencing mental health problems.

### 2.2. Family Carers

Invariably, most people with autism live with their families in affluent as well as less-affluent countries. Family carers also report mental health issues. A review of 45 studies from across the globe [8] found that parents of children with autism experienced higher levels of parenting stress than parents of typically developing children and, for some, their scores were above the clinical thresholds of severity. Stressed parents also had higher risks of depression and anxiety. Problem behaviours of the child and sensory issues were associated with increased parental stress: single young mothers with poor coping strategies were at greatest risk. Notably, some studies suggested that stressed parents also increased their child’s problem behaviours. It was a two-way relationship.

Parental stress continues when caring for adult persons with autism [9]. Unsurprisingly, a review of parental mental health during the COVID-19 epidemic found that the anxiety and stress of the parents increased alongside a need for more support compared to the pre-pandemic period [10].

Mental health issues in people with autism and their families has significant repercussions for them. Reviews and meta-analyses of studies report a significantly lower quality of life [11], increased risk of unemployment [12], sparser social networks, higher rates of loneliness [13], possibly greater risk of substance abuse [14] and higher incidence of self-harm and suicide [15]. Mental health problems cannot be dismissed as a consequence of autism—not least because a majority of people with autism escape having these issues—but those who have mental health problems need extra support to help them cope with and hopefully eliminate them.

### 2.3. Promoting Better Mental Health in People with Autism

Three major questions arise from this research on mental health and autism. They will be the focus of the remainder of this paper.
How best can autistic persons with mental health issues be treated?How can mental health problems be prevented?How can support services better respond to the needs of people with autism?

Table 2 summarises the key issues identified under each question. Although they are presented separately within the three columns, their impact is greater when they are implemented across the three domains of treatment, prevention and service development.

## 3. How Best Can Autistic Persons with Mental Health Issues Be Treated?

### 3.1. Improved Access

The conclusion of many studies noted above is that people with autism need improved access to the mental health services that are available to the general population. It is an obvious starting point given the struggle for equity in health services, the anti-discrimination legislation that is in place in many countries and the expertise and experience of the clinicians who work in such services. However, the reality across all nations is that mainstream services are often ill-equipped to treat children, youth or adults with autism. The reasons are manifold [16]. The services are already over-stretched and under-resourced, waiting times for assessment and treatment are long and staff lack experience in adapting their procedures to the needs of people with autism.

### 3.2. Autism Adapted

A review of 12 studies that investigated the barriers and facilitators of access to mental health services in affluent countries found that the most commonly reported barrier was a lack of therapist expertise in autism or an inability or unwillingness on their part to tailor their procedures to support autistic patients [17]. This and other short-comings are even more pertinent in rural communities [18] and in less-affluent countries with less developed or non-existent mental health provision [19]. Nonetheless, access to medication for children and adults mostly requires access to psychiatrists. While it can be beneficial for co-morbid conditions such as ADHD in children with autism, there are significantly increased risks of adverse effects and concerns around the long-term usage of anti-psychotics with children [20].

### 3.3. Fexibility

The experiences of persons with autism, family carers and health professionals who had experienced mental health services were captured in a review of 38 studies that used qualitative analysis from interviews and focus groups [21]. Overall, their experiences were predominantly negative. Three main themes commonly featured in the responses. They had a lonely, frustrating and difficult experience with the services. Their autism required the services to be flexible in adjusting to their communication and style of interaction, but this did not happen. A trusted relationship was needed with clinicians that empowered them to become more independent and socially active. The review authors also highlighted the over-reliance on medication in the mistaken belief that people with autism cannot engage in talk therapies. They also warned that the current systems potentially could cause more harm than good with the risk of worsening the individual’s condition through iatrogenesis. 

### 3.4. Family Functioning

Furthermore, the focus in mental health services is invariably on the individual and the impact on family carers and the functioning of the family is rarely addressed directly. Instead, the parent or sibling may have a separate referral to another clinician if they are experiencing mental health issues. Nonetheless, interventions that target the family system, rather than a focus on the individual, are a well-established and effective approach for a wide range of clinical needs [19]. A recent review of 11 studies with families who have a member with autism or other developmental disabilities reported positive effects on wellbeing and family relationships [22]. However, none addressed the mental health needs of either the person with autism or their carer.

### 3.5. Biopsychosocial Models

Increased investment in mental health services could be one solution and arguably is the one most likely to be advocated by clinicians working in such services. But other approaches must be considered given the additional complexities faced by people with autism and, indeed, with other co-morbid conditions such as intellectual disability, brain injury and dementia. The starting point is to reconsider whether to use a medical model of mental health—that sees it as an illness that needs to be cured—or whether it is better conceived as a biopsychosocial condition, as others have argued [23]. Indeed, this tripartite conception is equally pertinent to our fuller understanding of autism, which enhances its relevance for this population.

The central assumption behind the biopsychosocial model is the interdependence between biological (e.g., genetics, neurochemistry), psychological (e.g., self-esteem, social skills) and social factors (e.g., inter-personal relationships, lifestyle) in influencing a person’s health and wellbeing. This model underpinned the International Classification of Functioning, Disability and Health, (known more commonly as ICF) promoted by the World Health Organisation, which also emphasises the environmental influences (such as poverty and housing) on human development [24]. These conceptualisations broaden the range of supports to be considered in responding to the needs of people with autism and especially those experiencing mental ill-health. While there might be a case for creating specialised mental health services for people with autism, they will still need to be delivered within a more holistic support paradigm. Nevertheless, the risk is that investments in the latter will not take place. Moreover, in low-resourced countries, a focus on social and community responses to autism is likely to prove more cost-effective than clinical services. Section 5 will explore these options more fully.

## 4. How Can Mental Health Problems Be Prevented?

The medical model of mental health fails when it comes to prevention. Rather, the promotion of better mental health comes from enhancing the psychological, social and environmental influences on a person’s wellbeing. There is a growing body of evidence for their effectiveness for people with autism. For example, a British study found that the quality of life of adults with autism was greater for those in employment and who were receiving social supports and in a relationship [25]. Likewise, the majority of children and youth with autism do not experience mental health issues. So what has protected them from becoming overly anxious and depressed, for example? Thus far, insufficient studies have been undertaken to give a definitive answer, but we can get some clues from those that have been undertaken. Here are the five most promising strategies and although they are presented separately here, their power is increased when they are combined.

### 4.1. Encouraging Social Connections, Developing Friendships and Reducing Social Isolation

This begins within families, as the children are encouraged to join in family events at home and on visits to relatives. These can develop into attending community activities such as walks, swimming and cinema (in quieter locations and at quieter times). Opportunities to interact with their peers can be gradually introduced through play-dates at home or through joining in sports clubs or other leisure activities. The school years bring further opportunities for friendships to be nurtured, although these may need to be facilitated by teachers, leaders of community groups or support staff. The social connections and friendships need to be maintained through the transition to adulthood, which is a good reason to continue or extend membership of community clubs. All of this inculcates a sense of belonging that seems to be a key element in human quality of life [26]. A review of 22 studies that investigated the issue of friendships by listening to the experiences of mostly youth with autism identified what friendships meant to them and their experiences of them, including the challenges and fears they had encountered [27].

### 4.2. Increasing Social Motivation

A common feature of autism is the avoidance of social interactions with others. But low social motivation, as it has been termed, brings with it an increased chance of loneliness that in turn leads to depression with reduced participation in education or seeking employment [13,28]. Increased motivation comes from experiencing enjoyable interactions with others, so the connections noted above may have to be carefully graded or adjusted to ensure that they do not become aversive. The presence of a trusted partner (such as a parent or sibling) can be helpful even if they remain in the background. Being part of a group of teenagers and young adults with autism also builds motivation as they meet like-minded individuals who may be more tolerant of their unusual traits as well as providing role models for appropriate social behaviours [29].

### 4.3. Increased Autonomy and Self-Determination

A third strategy is to encourage the person’s capacity to self-regulate their feelings. This begins and is sustained by family members and support staff observing and listening to the child’s non-verbal and verbal communications. For example, there will be signs that the child or teenager is building up to a “melt-down” and when they are detected, the parent can respond by confirming the child’s feelings and suggesting actions that would help to calm the child. When the child is relaxed, the situation can be reviewed and the actions that helped the child can be identified. A growing number of studies have also reported the value of mindfulness training for older children and adults, which also encourages self-control of one’s feelings [30]. Cognitive behavioural training (CBT) is also emerging as a helpful intervention with young autistic adults, primarily male, without an intellectual disability [31]. More broadly, the young people are encouraged to make decisions for themselves—starting with their choice of clothing and food—but developing their autonomy by increasing their independence beyond the family through choosing what they do, with whom they do it and the “rules” to be followed, principally to keep them safe [32]. From an early stage, parents and staff should adopt a response interactive style rather than giving directions and demands that keeps them in a controlling role rather than ceding that to the person with autism [33].

### 4.4. Build on The Person’s Assets, Strengths and Talents

A diagnosis of autism can create a mindset that focusses on the persons’ deficits and weaknesses with the associated assumption that these must be remedied. Positive psychology advocates a different approach: one that builds on the strengths and talents of the person. Various psychosocial interventions have used this thinking to reduce mental health disorders and to promote mental wellbeing, but their use within autism has been limited. A review of 24 studies that used this approach informed the development of a conceptual model for creating strength-based interventions [34]. Four categories of strengths were identified among people with autism: perceptual (e.g., attention to detail, memory); reasoning (e.g., logical reasoning; problem solving); expertise (e.g., special interests; in-depth knowledge); character strengths (e.g., honesty; creativity). Once an individual’s strengths are identified through self-reports and from the family allied with observations from support staff, an intervention plan can be developed—such as those noted above—that incorporates their interests and talents. To date, the efficacy of this approach remains uncharted, although given the heterogeneity within the autistic population this will have to focus on individual rather than group outcomes. At worst, though, a strength-based approach will likely produce less stress compared to one that focuses on deficits and risks despair when no improvement occurs.

Families and service personnel will have found other strategies that have worked for them, albeit with specific individuals, hence the value of “peer-to-peer” learning, such as parent groups, mentoring or “in-house” training sessions for staff. The over-riding aim is to enable people with autism to enjoy the same opportunities as their neurotypical peers in education and training, obtaining paid employment, forming romantic relationships and having their own accommodation—in sum, to enjoy an equivalent quality of life, which is not the case for many at present.

### 4.5. Supporting Parents

The family’s quality of life also needs to be addressed, as this will invariably impact on the person with autism. As noted above, family-system interventions hold promise for achieving this goal and have been proposed as the bedrock on which child development interventions should be built [35]. As yet, research aimed at improving the quality of life of families who have a member with autism is sparse, although a recent review of 17 studies concluded that most family-focussed interventions had been effective [36]. These took various forms: home-based guidance from a therapist, peer-mentoring from other parents, family-centred activities, group training workshops and support groups. The formation of a trusted, personal relationship between the mothers and the support workers involved with their child seems an essential requirement in helping mothers especially in terms of their emotional wellbeing. A review and meta-analysis of 17 studies that aimed to improve the mental wellbeing of mothers of children with disabilities found that cognitive-behavioural and psycho-educational interventions led to reduced stress and/or better maternal health [37]. But the small number of studies and limited sample sizes mean that their usefulness with most parents is untested.

Most of the studies reviewed in this section were conducted in affluent countries. Nevertheless, they do offer insights that can be replicated in less-affluent nations, as these preventative approaches are not heavily reliant on having a highly skilled, professional workforce and can be implemented at relatively low cost [38]. Community staff and family members are the main deliverers of the interventions, guided and advised by managers with some expertise in autism, as the next section proposes.

## 5. How Can Support Services Better Respond to the Needs of People with Autism?

The reviews identified above often proposed various conceptual models for responding to the mental health needs of people with autism. Often, these models were informed by the research they had reviewed but they also drew on theoretical perspectives relating to autism and/or mental health. In this section, five common features across the different conceptual models are highlighted with the aim of implementing more effective support services for people with autism that promote positive mental health.

### 5.1. Tiered Services

A common response to meeting the needs of specific groups such as people with autism has been to provide specialist services with dedicated clinicians or educators. Special schools or centres are examples of such services, as are teams of professionals working from clinics or hospitals who provide assessment and treatments on a time-limited basis. Such services are costly to maintain and often the demand for these services exceeds their capacity, with the result that waiting lists build up because of too many referrals. One solution is to ensure that the specialist services are confined to people with more complex needs who require the skills of highly experienced clinicians. But alternative and additional services then need to be resourced for persons with less complex needs, which require additional funding or, more likely, transferring some resources out of specialised provision.

A tiered approach has been proposed for children’s healthcare services [39], which could be adapted to meet the needs of those with autism. The first tier involves equipping the primary care services such as community doctors and nurses to recognise the signs of autism and offer first-line advice as well as sign-posting families to counselling and community services, such as preschool centres, within their locality. A second-tier service might consist of existing community-based services—such as social workers, speech and language therapists or occupational therapy—for further advice and guidance. A third-tier service could be created that focuses on family support for children with autism, or for adult persons. These could be staffed by “mid-level” workers such as paraprofessionals with training in autism who are able to provide home-based and community support over an extended time period if required [40]. The fourth tier would be the specialist autism service with boosted expertise in managing mental health issues for the person with autism and family members. It is crucial that the pathways across these tiers be charted, to clarify referral criteria and procedures, for example. The fifth tier arguably would be the procedures to support cross-sectoral working bringing together health, social care, housing, employment and leisure services, as will be mentioned later. In planning tiered services for the prospective population of persons with autism within their designated service area, an estimate of the likely number of people who could avail of services at each level could be based on the profile of past and current service users and available assessment and intervention records allied with likely prevalences reported in the literature.

### 5.2. Training of Staff 

Mechanisms for sharing knowledge and skills about autism is a vital component in creating better services. Here, too, training can adopt a tiered approach. Raising public awareness about autism is the first tier, not least as it will help parents, the wider family circle and also staff working in child care, preschool centres and schools recognise signs of autism in a toddler. Existing professionals in primary care or mainstream services—a second tier—also need to have a better understanding of how people with autism can be helped and the available supports to which parents can be directed [41]. Third-tier training would aim to equip staff working in community and home-based services for people with autism to undertake assessments, devise individual support programs, encourage participation in community activities, develop social connections and provide practical guidance and emotional support to family carers. Staff working in fourth-tier services would need to augment their professional training with in-depth and more technical autism training around diagnosis, autism-adapted treatments and family-system interventions set within a transdisciplinary approach.

Fortunately, a variety of different training approaches and methods can be used in all these different tiers—face-to-face workshops, online courses, video programmes, training manuals, practical exercises, support groups and one-on-one coaching [42]. One essential, though, is to have a tutor or teacher who has practical experience and knowledge about autism, excellent communication skills and an outgoing personality to deliver or oversee the training that is being delivered. Such trainers are needed in all the service tiers, with those in higher tiers also acting as the trainers of persons in the lower service tiers, the so-called “training-of-trainers” approach.

### 5.3. Co-Production with Autistic Advocates

Within health and social care services, traditionally it has been the professionals—clinicians and administrators—who have designed and delivered the services. This model works best when “expert knowledge” is required to make a diagnosis of physical illness or to perform surgery, but for conditions like mental illness and autism, the experts are also the person with the condition and their family carers; hence, the term “advocates”. Modern practice now emphasises the involvement of current and potential service users in the co-production or modification of existing services and the design of new ones [43]. It is seen as essential in producing effective services for persons with autism [19] and shifting the focus away from an illness orientation to one that promotes emotional wellbeing. According to a recent review [44], autistic people’s perspectives are essential to success in planning interventions for individuals, but this ethos must be extended to wider service planning and its implementation, most notably as trainers of staff who may be working in different tiers of service and devising their job specification. This may lead to advocates acting as peer mentors [45] or obtaining paid employment within newly created services. In a sense, there is no shortage of trainers when there are people with autism and their families available to take on this role.

### 5.4. Mobilising Community Resources

The biopsychosocial model described earlier demands a re-evaluation of the range of supports and interventions required to provide a better quality of life for persons with autism—especially those with mental health issues—and for their family carers who continue to look after their relative into adulthood [25,44,46]. Modern services are slowly shifting from clinic-based health treatments towards a model of holistic working to improve the quality of people’s lives grounded on an ethos of human rights and equal opportunities [47]. This means ensuring people with autism have equitable access to all the community resources such as further education and training, housing, paid employment, public transport as well as social and leisure pursuits such as sports and the creative arts [48]. At the individual level, it may be a matter of personal approaches to the relevant agencies, but better still if efforts are made to enhance cross-sectoral working at a local if not national level so many more people can benefit. The so-called “silo mentality” that is present in many government bureaucracies is the biggest inhibitor of co-ordinated action across sectors and, as yet, we have few examples in the field of disability and mental health as to how best it can be achieved. It is likely, though, that visionary leadership by an individual or a small group of leaders will be an essential ingredient to success [49].

### 5.5. Cultural Adaptations

It should be obvious that social interventions of the sort proposed for people with autism need to take full account of the culture into which they have been planted, but this is often ignored. Services models are shared across affluent countries, which admittedly often share a common language and culture, but then these models are imported into countries with very different cultures. However, the effectiveness of these models is blunted at best. A recently published conceptual framework charts the cultural and contextual influences that need to be taken into account in relation to autism and its identification, diagnosis and interventions [50]. The authors describe cultural norms of typical and atypical behaviours, culture-specific approaches to parenting, mental health literacy, cultural beliefs, attitudes and stigma to disability, as well as the need to reflect on the affordability, availability, accessibility, and acceptability of services in low- and middle-income countries (LMICs) in particular. Indeed, these insights must be borne in mind when responding to the needs of people from minority cultures living in affluent countries as citizens or immigrants [51].

## 6. Concluding Comments

This review is limited in a number of respects. It draws only on the published literature from academic journals. Further and probably richer insights could be gained from the experiences of practitioners across the globe who have created innovative supports for people with autism in their locality. But often, busy practitioners do not have the time to write about their experiences. Perhaps this paper will nudge them to make time to tell others—perhaps even train others—to emulate their services.

A further weakness is that most of the research studies and, hence, the identified reviews focus on more affluent, English-speaking countries, which also describes the author’s culture, although I can claim a long involvement with colleagues working in LMICs. The danger is that we fail to recognise that our views of autism and mental health are also culture-specific as we laud the advances made to understand and respond to these conditions. The implicit devaluing of alternative concepts and approaches can impede the emergence of more culturally appropriate supports and intervention, as a review of reviews that contrasted autism interventions in high- and low-income countries confirmed [52]. More efforts need to be expended on research and development in LMICs, especially in relation to emerging issues such as mental health and autism. Table A1 documents the dearth of studies from LMICs in the past decade.

Every review ends by calling for further research and this is no exception. Yes, we could seek to recruit larger samples, devise better measurement tools, run more control trials, use more sophisticated statistical and qualitative analyses and write even more papers in peer-reviewed journals. My fear, though, is that such a quest becomes a cul-de-sac in which addressing the present needs of people with autism and mental health in the villages, towns and cities across the globe becomes secondary. Rather, research resources would be better spent on increasing our knowledge as to how new forms of effective and efficient support services can be created and sustained in even the most impoverished countries. Identifying the knowledge and skill requirements of the managers and staff of such services will be an essential component in their newly created job descriptions, as will be strategies for promoting the advocacy and self-determination of people with autism and family carers. Such transformations will not be easily achieved, but we can take heart that pockets of it are emerging across the world that give us a better sense of what is attainable. The coming decades will test whether the seemingly impossible can become possible.

## Figures and Tables

**Table 1 brainsci-13-01645-t001:** Prevalence rates (with 95% confidence intervals [CI]) for mental health problems in autism samples and the general population.

Mental Health Problem	Autism Sample	General Population
Attention-deficit hyperactivity disorder	28% (CI 25–32%)	7.2% (CI 6.7 to 7.8%)
Anxiety disorders	20% (CI 17–23%)	7.3% (CI 4.8–10.9%)
Disruptive, impulse-control and conduct disorders	12% (CI 10–15%)	7.0% (CI 4.0–10%)
Depressive disorders	11% (CI 9–13%)	4.7% (CI 4.4–5.0%)
Obsessive–compulsive and related disorders	9% (CI 7–10%)	0.7% (CI 0.4–1.1%)

Source: Zeidan et al. (2022) [7].

**Table 2 brainsci-13-01645-t002:** Key issues identified.

Treatment	Prevention	Service Development
Improved access	Social connections	Tiered services
Autism adapted	Social motivation	Training for staff
Flexibility	Self-determination	Co-production
Family carers	Personal assets	Community resources
Biopsychosocial approaches	Supporting family carers	Cultural adaptations

## Data Availability

All the reviews cited in this paper are publicly available and links are provided to enable readers to access them.

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
