# Peer review of "Nurturing the Positive Mental Health of Autistic Children, Adolescents and Adults alongside That of Their Family Care-Givers: A Review of Reviews"

_brainsci, 2023, doi:10.3390/brainsci13121645_

Round 1

Reviewer 1 Report

Comments and Suggestions for Authors

The scoping review looks at mental health in ASD and proposed mental health services. 

My major concern is that the review is not clearly answering
a) how can mental health problems (in persons diagnosed with an autism spectrum disorder) be prevented - that needs to be better carved out
b) how best can persons with an autism spectrum disorder (it is incorrect to talk about autistic persons, please correct this throughout the manuscript!!!) with mental health issues be treated - also here the review stays at the health service level - this is either because the aim is to propose the 4 or 5-tier system and its efficiency or due to the variability and flexibility required for each person. This is not made clear.

In general, the manuscript would benefit it the collated agreement from the various reviews is summarized in a table, i.e. the 4-tier system for providing health service, and crucially for whom it is, what would be the expected number of persons to treat, so that stakeholders can use this information to dimension the service appropriately.

This would at least clearly answer the third question, namely how can support services better respond to the needs of persons and caretakers of persons with an autism spectrum disorder. 

a few sentences / section read a bit strange

1. Unlike other disabilities – physical, sensorial and cognitive - it has only recently been recognised as a distinct condition.

not sure what you mean by recently, but ASD is a distinct condition since DSM-III.

2. Nonetheless access to medication for children and adults mostly requires access to psychiatrists. While it can be beneficial for conditions such as  ADHD in children, there are significantly increased risks of adverse effects and concerns around the long-term usage of anti-psychotics [20]

your article is about ASD, not SCZ, even if some get anti-psychotics, this is not the most common treatment for ASD

minor:

line 118 metal <- should be mental

line 155 that most commonly reported barrier was <- "the" missing between that and most

line 174 through iatrogenic harm <- better to write "through iatrogenesis" (tautology to say iatrogenic harm)

line 330 first Tier involves <- should be tier

line 389 are slowing shifting from clinic-based health treatments towards  <- should be slowly

line 391 grounded an ethos of human rights and equal <- on missing, i.e. grounded on an ethos ...

line 413 in LMICs <- abbreviation not established (is it low middle income countries?)

Author Response

My major concern is that the review is not clearly answering
a) how can mental health problems (in persons diagnosed with an autism spectrum disorder) be prevented - that needs to be better carved out

b) how best can persons with an autism spectrum disorder (it is incorrect to talk about autistic persons, please correct this throughout the manuscript!!!) with mental health issues be treated - also here the review stays at the health service level - this is either because the aim is to propose the 4 or 5-tier system and its efficiency or due to the variability and flexibility required for each person. This is not made clear.

In general, the manuscript would benefit it the collated agreement from the various reviews is summarized in a table, i.e. the 4-tier system for providing health service, and crucially for whom it is, what would be the expected number of persons to treat, so that stakeholders can use this information to dimension the service appropriately. This would at least clearly answer the third question, namely how can support services better respond to the needs of persons and caretakers of persons with an autism spectrum disorder. 

Many thanks for your careful reading of the paper and especially for your helpful comment to carve out more clearly the issues identified in the reviews.  To that end I have taken on board your suggestion of including a summary table which I have done at the end of section 2 as an overview of issues identified in response to the three questions while also noting their applicability across the three domains of treatment, prevention and service developments.

I was cautious about making one issue more important than another; such as tiered services on which the reviewer had focused.  As the reviewer noted, not only different tiers of service are needed but other pertinent issues also require  attention, due to the variability among persons with autism; the flexibility needed in responding to individual need and the varying capacity of services to meet certain needs. 

With respect to your comment around terminology, you are probably aware of the growing proportion of self-advocates who now prefer the term 'autistic', so I have used the term inter-changeably with 'persons with autism'.  Nevertheless, I appreciate that it might be more consistent to use one term and will take advice from the technical editors in that regard.

You rightly note the need for more detailed planning of services within each tier,  but existing reviews do not provide guidance for doing this.  However I have noted that: "In planning tiered services for the prospective population of persons with autism within their designated services,  an estimate of the likely number of people who could avail of services at each level could be based on the profile of past and current service-users, available assessment and intervention records allied with likely prevalences as reported in the literature".  

A few sentences / section read a bit strange

  1. Unlike other disabilities – physical, sensorial and cognitive - it has only recently been recognised as a distinct condition.  not sure what you mean by recently, but ASD is a distinct condition since DSM-III. 

This has been changed to read: "Unlike other widely known disabilities – physical, sensorial and cognitive - it was only in 1980 that autism was clinically recognised as a distinct condition".  

2. Nonetheless access to medication for children and adults mostly requires access to psychiatrists. While it can be beneficial for conditions such as  ADHD in children, there are significantly increased risks of adverse effects and concerns around the long-term usage of anti-psychotics [20]  your article is about ASD, not SCZ, even if some get anti-psychotics, this is not the most common treatment for ASD.

This has been changed to read:  "While it can be beneficial for co-morbid conditions such as ADHD in children with autism, there are significantly increased risks of adverse effects and concerns around the long-term usage of anti-psychotics with children [20]".

minor:  Apologies for the typos which have been corrected.

Reviewer 2 Report

Comments and Suggestions for Authors

Thank you very much for the possibility to review this Article. The Topic of the Paper is interesting. 

Unfortunately, I have a problem with the Structure of the Article:

The Author declares that this is the Scoping review. But I didn't find the basic information, that is usually used in Scoping review - articles:

1. Protocol of Scoping review - whether it has been published

2. Scoping question + PCC strategy 

3. inclusive criteria

4. exclusive criteria 

5. list of databases that were searched

6.  flow chart - the procedure of elimination of the Articles 

The article is quite a good analysis of the problem, but I'm missing basic information on how the author got to this information.

In this form, I cannot recommend the article for publication. If it is to be a scoping review, it is necessary that the above information be added.

Author Response

Thank you very much for the possibility to review this Article. The Topic of the Paper is interesting.  The article is quite a good analysis of the problem, but I'm missing basic information on how the author got to this information.

Many thanks for reviewing the article and the time you devoted to it.  Your ratings of 4 or 5 stars to all but one of the ratings is much appreciated. 

Unfortunately, I have a problem with the Structure of the Article:

The Author declares that this is the Scoping review. But I didn't find the basic information, that is usually used in Scoping review - articles:

When undertaking the study, I debated with myself as to whether or not it was sufficiently robust to be considered a scoping review.  In light of your comments, I have reverted to it being a literature review but still retaining the scoping questions, the description of the search terms and the databases used.  I have also noted additional limitations arising from this study and the possible guidance it offers for further scoping or systematic reviews.    Hence all references to scoping review have been deleted.  

For example, the introduction now states: "A literature review of the reviews was then undertaken with the primary aim of identifying recommendations for practice based on the insights and evidence across the diversity topics affecting the mental health of persons with autism".  

Also a further caution has been added:   "Moreover this literature review could usefully guide future scoping or systematic reviews on the topic of mental health and autism which would help to overcome any subjective biases of this author [1]"  The added reference is: 

  1. Munn, Z., Peters, M.D.J., Stern, C. et al. Systematic review or scoping review? Guidance for authors when choosing between a systematic or scoping review approach. BMC Med Res Methodol 18, 143 (2018). https://doi.org/10.1186/s12874-018-0611-x

I had also noted two further considerations when undertaking the review: namely guidance for practitioners and insights for low and middle income countries, as the paper was prepared in response to an invitation to contribute to a special issue on that topic. 

Further limitations of the study are given in the Conclusion: lines 439 to 455. 

Round 2

Reviewer 1 Report

Comments and Suggestions for Authors

Thank you very much for revising your manuscript. Table 2 (note that it is in our manuscript table 1) and the subheadings in section 3 do help the reader and stressing your points.

minor issues

line 56 "would help to overcome any subjective biases of this author [1]" <- should be the author(s) 

line 280 "assets,strengths and talents" <- space missing after assets,

Reviewer 2 Report

Comments and Suggestions for Authors

Dear Author,

I am happy to recommend it for publication.

Thank you for understanding and accepting the difference between scoping and literature review.